# Leveraging expression from multiple tissues using sparse canonical correlation analysis and aggregate tests improves the power of transcriptome-wide association studies

**Helian Feng**[1,2]*, **Nicholas Mancuso**[3,4], **Alexander Gusev**[5,6,7], **Arunabha Majumdar**[8,9], **Megan Major**[10], **Bogdan Pasaniuc**[8,9], **Peter Kraft**[1,2]

**1** Department of Epidemiology, Harvard T.H. Chan School of Public Health, Boston, Massachusetts, United States of America, **2** Department of Biostatistics, Harvard T.H. Chan School of Public Health, Boston, Massachusetts, United States of America, **3** Center for Genetic Epidemiology, Department of Preventive Medicine, Keck School of Medicine, University of Southern California, Los Angeles, California, United States of America, **4** Division of Biostatistics, Department of Preventive Medicine, Keck School of Medicine, University of Southern California, Los Angeles, California, United States of America, **5** Department of Medical Oncology, Dana-Farber Cancer Institute & Harvard Medical School, Boston, Massachusetts, United States of America, **6** Division of Genetics, Brigham & Women's Hospital, Boston, MA, United States of America, **7** Medical and Population Genetics, Broad Institute, Cambridge, Massachusetts, United States of America, **8** Department of Human Genetics, University of California Los Angeles, Los Angeles, California, United States of America, **9** Department of Pathology and Laboratory Medicine, University of California Los Angeles, Los Angeles, California, United States of America, **10** Bioinformatics Interdepartmental Program, University of California Los Angeles, Los Angeles, California, United States of America

* helian@g.harvard.edu

## Abstract

Transcriptome-wide association studies (TWAS) test the association between traits and genetically predicted gene expression levels. The power of a TWAS depends in part on the strength of the correlation between a genetic predictor of gene expression and the causally relevant gene expression values. Consequently, TWAS power can be low when expression quantitative trait locus (eQTL) data used to train the genetic predictors have small sample sizes, or when data from causally relevant tissues are not available. Here, we propose to address these issues by integrating multiple tissues in the TWAS using sparse canonical correlation analysis (sCCA). We show that sCCA-TWAS combined with single-tissue TWAS using an aggregate Cauchy association test (ACAT) outperforms traditional single-tissue TWAS. In empirically motivated simulations, the sCCA+ACAT approach yielded the highest power to detect a gene associated with phenotype, even when expression in the causal tissue was not directly measured, while controlling the Type I error when there is no association between gene expression and phenotype. For example, when gene expression explains 2% of the variability in outcome, and the GWAS sample size is 20,000, the average power difference between the ACAT combined test of sCCA features and single-tissue, versus single-tissue combined with Generalized Berk-Jones (GBJ) method, single-tissue combined with S-MultiXcan, UTMOST, or summarizing cross-tissue expression patterns using Principal Component Analysis (PCA) approaches was 5%, 8%, 5% and 38%, respectively. The gain in power is likely due to sCCA cross-tissue features being more likely to be

**Data Availability Statement:** Precomputed sCCA weights for GTEx v6 and v8 are available at: http://gusevlab.org/projects/fusion/#reference-functional-

data. Code to construct sCCA weights and perform sCCA TWAS is available at: https://github.com/fenghelian/sCCA-ACAT_TWAS.

**Funding:** This work was supported by grants from the U.S. National Institutes of Health: U01 CA194393, R01 HG009120 (BP), R01 CA227237 (AG), R35 CA197449 and U01 HG009088. The funders had no role in study design, data collection and analysis, decision to publish, or preparation of the manuscript.

**Competing interests:** The authors have declared that no competing interests exist.

detectably heritable. When applied to publicly available summary statistics from 10 complex traits, the sCCA+ACAT test was able to increase the number of testable genes and identify on average an additional 400 additional gene-trait associations that single-trait TWAS missed. Our results suggest that aggregating eQTL data across multiple tissues using sCCA can improve the sensitivity of TWAS while controlling for the false positive rate.

## Author summary

Transcriptome-wide association studies (TWAS) can improve the statistical power of genetic association studies by leveraging the relationship between genetically predicted transcript expression levels and an outcome. We propose a new TWAS pipeline that integrates data on the genetic regulation of expression levels across multiple tissues. We generate cross-tissue expression features using sparse canonical correlation analysis and then combine evidence for expression-outcome association across cross- and single-tissue features using the aggregate Cauchy association test. We show that this approach has substantially higher power than traditional single-tissue TWAS methods. Application of these methods to publicly available summary statistics for ten complex traits also identifies associations missed by single-tissue methods.

## Introduction

Genome-wide association studies (GWASs) have successfully identified thousands of associations between single-nucleotide polymorphisms (SNPs) and complex human phenotypes. Yet, the interpretation of these identified associations remains challenging, and several lines of evidence suggest that many additional associated loci remain to be identified [1,2]. A recently proposed approach transcriptome-wide association study (TWAS) [3,4] identifies genetic associations by combining GWAS data with expression quantitative trait locus (eQTL) data. TWAS can be used both to identify new associations and prioritize candidate causal genes in GWAS-identified regions [5]. TWAS integrates gene expression with GWAS data using only genotype expression imputation from a gene expression model built from eQTLs and then tests for the association between imputed gene expression level and a phenotype of interest. The main strength of TWAS is that it can infer the association of imputed gene expression with the phenotype using only GWAS summary statistics data [3,4]. TWAS can increase the statistical power by combining single-SNP association tests in a biologically motivated fashion and reducing the number of tests performed. The applications of TWAS have led to novel insights into the genetic basis for several phenotype and diseases [6].

Despite the successes of TWAS, the approach has multiple limitations [7]. First, the most relevant tissue for many human diseases and phenotypes remains unclear, and the eQTL data for these relevant tissues are usually challenging to access in large samples. The choice of the most relevant tissue-specific eQTL sample for building gene expression prediction model in TWAS remains largely ad-hoc. Two commonly adopted approaches are: (1) using the largest eQTL sample accessible (usually whole blood [3]), or (2) using the most relevant tissue based on previous knowledge and experience [6,8]. Second, the power of TWAS is mainly bounded by the sample size of eQTL data; power of TWAS increases dramatically with the eQTL sample size, approaching an empirical maximum when eQTL sample size is close to 1,000 [3]. However, most available eQTL data sets have a sample size substantially smaller than 1,000. For

example, Genotype-Tissue Expression(GTEx) project [9,10] have generated matched genotype and expression data for 44 human tissues, but with sample size for each tissue varying from only 70 to 361. Researchers do not always know which tissue to use, and sometimes the sample size for the tissue that they prefer to use is too small to have enough power.

Recent work in gene regulation patterns across tissues suggests that local gene expression regulation is often shared across tissues [9–11]. Thus, combining eQTL data across multiple tissues can improve the power of TWAS, by increasing the effective eQTL sample size or increasing the likelihood that the causal tissue (or a close proxy) is included in the eQTL training data. Two previously proposed approaches, UTMOST [12] and S-MultiXcan [13], have shown the advantage of a multi-tissue TWAS approach. However, these two approaches still conduct the TWAS test with single-tissue TWAS weights first, and then combine multiple single-tissue associations into a single powerful metric to quantify. UTMOST uses a generalized Berk-Jones (GBJ) test, which is a set-based method [12]. S-MultiXcan proposes a combined chi-square test that uses principal components from the tissue-specific genetically predicted expression values to integrate univariate S-PrediXcan results [13]. We refer to these two approaches as single-tissue based cross-tissue TWAS approach. We propose to leverage the correlated gene expression pattern across tissues in the eQTL dataset directly to build more stable and representative cross-tissue gene expression features using sparse canonical correlation analysis (sCCA) [14], and thus improve the gene expression prediction model for TWAS. The potential advantage of sCCA is that it can capture any genetic contribution to gene expression that is shared across multiple tissues. Because sCCA maximizes the correlation between a linear combination of tissue-specific expression values and linear combination of cis-genotypes, sCCA features are more likely to be detectably heritable than cross-tissue features constructed using principal components analysis (PCA), which constructs linear combinations to capture total (genetic plus non-genetic) expression variance [14]. In addition, we also propose an omnibus test that combines the single tissue TWAS test results with the sCCA-TWAS test results using the aggregate Cauchy association test (ACAT). ACAT is a computationally efficient P-value combination method for boosting the power in sequencing study, and has proved to be powerful for detecting a sparse signal [15].

Specifically, we propose a novel four-step pipeline to perform multi-tissue TWAS: 1. generate sparse canonical correlation analysis (sCCA) [14] -based cross-tissue features (sCCA-features) integrating eQTL data across multiple tissues; 2. fit TWAS weights for these sCCA-features as well as single tissue-specific gene expression [3,4]; 3. perform TWAS with weights built from sCCA-features and singe tissue gene expression [3,4]; 4. combine the test results of sCCA TWAS results and single tissue TWAS results using the aggregated Cauchy association test (ACAT) [15]. We use extensive simulations to compare this approach with four other cross-tissue approaches, including: 1. performing TWAS on single most relevant tissue, 2. performing TWAS on all single tissues available and combining the test results via Bonferroni or generalized Berk-Jones (GBJ) test [16]; 3. using Principal Components Analysis (PCA) to create cross-tissue features; and 4. the recently proposed S-MultiXcan and UTMOST approach [12,13].

Through simulations we show that sCCA-features identify a larger number of cis-heritable transcripts than single tissue and PCA-features, and the combined test substantially improves statistical power. Importantly, all approaches successfully control the type I error rate. We also show by simulations that the power of our combined test compares favorably to other approaches despite using incomplete gene expression matrix for all individuals and all tissues thus requiring imputation, as is often the case for multi-tissue gene expression dataset like GTEx [9,10].

We applied our four-step approach to eQTL data from GTEx and 10 sets of publicly available GWAS summary statistics data. We built sCCA-features on an expression matrix including 134 individuals with data in 22 tissues. The sCCA-TWAS results were then compared with the single-tissue based TWAS results available on TWAS HUB (http://twas-hub.org). sCCA +ACAT TWAS was able to increase the number of testable genes by 81% and almost double the number of identified gene-phenotype associations (75% more genes identified).

The sCCA cross-tissue weights on GTEx version 6 and 8 are available on TWAS-HUB[17] and the Rscript to perform ACAT is also available at Github repository (https://github.com/yaowuliu/ACAT). The sCCA-TWAS could be easily performed with sCCA cross-tissue TWAS weights as traditional single-tissue TWAS and the test combination with ACAT is also easy to conduct. Sample code to compute sCCA cross-tissue weights and conduct sCCA+ACAT TWAS can be found at the Github repository (https://github.com/fenghelian/sCCA-ACAT_TWAS).

## Results

### Methods overview

We proposed sCCA+ACAT approach to conduct cross-tissue TWAS. Our proposed method entails four steps: the feature generating step, weight building step, TWAS step, and tests combining step (Fig 1). In the feature generating step, we considered two types of features: (1) single-tissue features which take the gene expression in each tissue as a separate feature; (2) cross-tissue features constructed as weighted averages of gene expression across tissues, where the weights were chosen using sCCA (see Methods for details)[14]. These weighted averages maximize the correlation between the weighted average of gene expressions across tissues and linear combinations of *cis*-genotypes (within 500kb of the gene boundary). In the weight building step, we build TWAS weights for each of these gene expression features by regressing the feature on cis-SNPs in the gene's window. In the second step of TWAS, we perform tests for association using these set of weights (for each single-tissue or multi-tissue feature) separately. Finally, in the tests combining step, propose a combined test of single-tissue test and sCCA cross-tissue test by combining the test results with ACAT [15].

We compared the performance of generating cross-tissue gene expression features with sCCA and PCA (Top 3 PCs on the gene expression) and the sCCA+ACAT TWAS results to

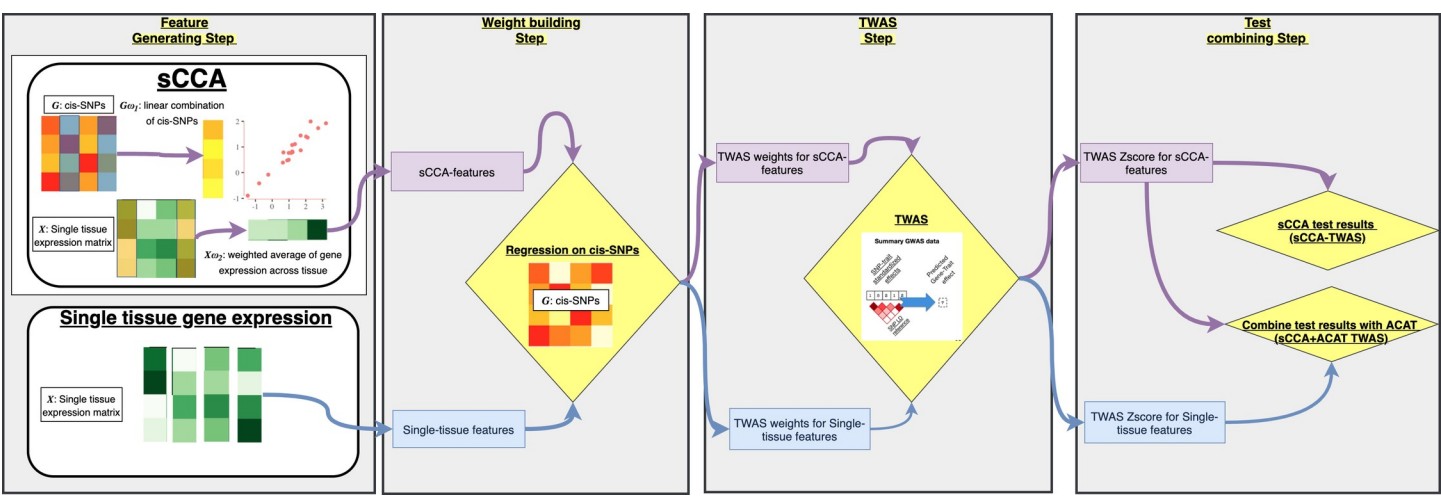

**Fig 1. Methods overview.** The single-tissue-based cross-tissue TWAS approach is shown in blue arrows, the PCA based cross-tissue TWAS approach is shown in red arrows, and the sCCA-TWAS approach is shown in purple arrows.

single-tissue based TWAS tests which combines the single-tissue TWAS test results using a Bonferroni multiple testing adjustment, the Generalized Berk-Jones (GBJ) procedure, or S-MultiXcan and UTMOST [12,13,16] (see Methods for more details) through 2,000 simulations based on GTEx data. We conducted the simulations varying gene expression heritability, genetic correlation in expression across tissues, the proportion of tissues correlated with the causal tissue, the scale of non-centrality parameters in the GWAS z-score distribution (to model GWAS sample size), and whether gene expression from the underlying causal tissue is observed (i.e. not included in model training) or not.

## sCCA improves statistical power to detect heritable gene expression

The first step of the TWAS approaches we consider tests the cis-heritability of each gene expression feature; only the features that demonstrate significant heritability are analyzed further. Fig 2 compares the power of this heritability test for single-tissue, PC and sCCA expression features in the scenario where half of the tissues are correlated with the causal tissue, and the causal tissue is not observed. The relative performance of these features is very similar in the other scenarios (S1 and S2 Figs). The power of detecting heritable genes at a set alpha level increases as the correlation between correlated tissue and causal tissue or the heritability for

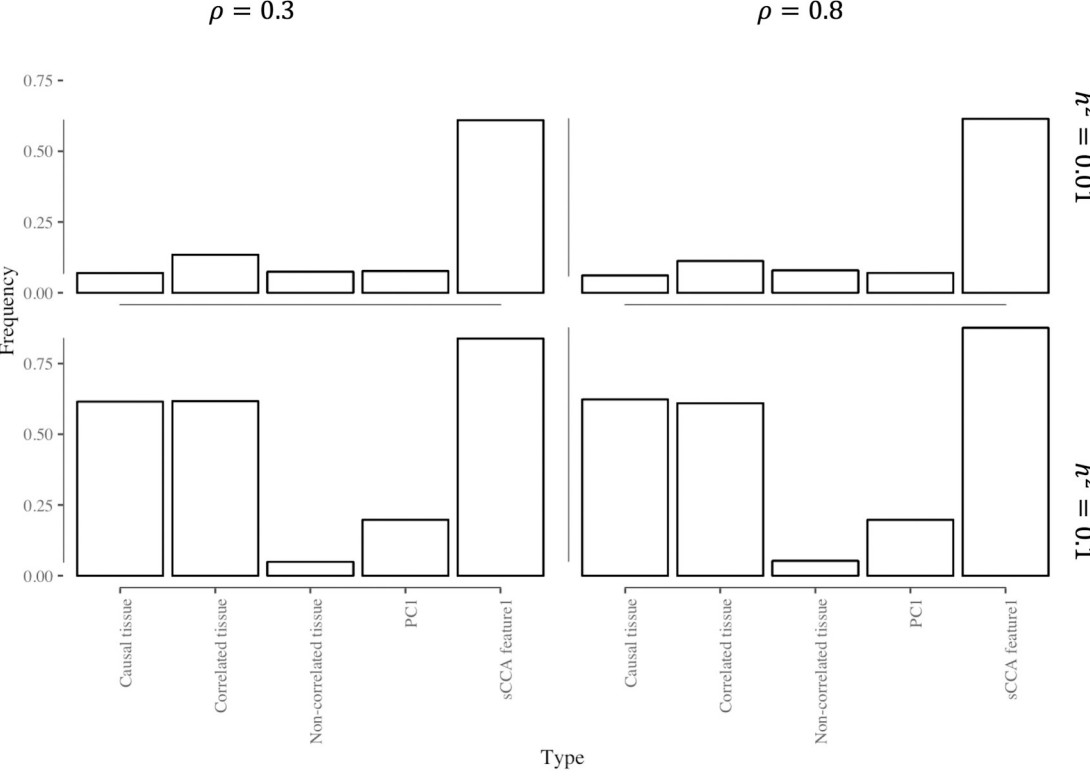

**Fig 2. Proportion of significant (p<0.05) heritability tests for different expression features when cis genetic variation is associated with expression in *some* tissues.** Here $\rho$ denotes the strength of the genetic correlation between expression in the causal tissue and another tissue in which the expression is also associated with cis-germline variation ("correlated tissues"). "Non-correlated tissues" are the tissues where local germline variation is not associated with the gene expression. Here expression in half of the tissues is genetically correlated with that in the causal tissue, and the causal tissue is *not* observed (performance in the causal tissue is included as a reference). PC1 is the first principal component of cross-tissue gene expression; sCCA-feature1 is the linear combination of tissue expression values from the first pair of sCCA canonical variables. $h^2$ denotes the proportion of expression variance in the causal tissue explained by cis-genetic variation.

gene expression in causal tissue increases. On average, the sCCA-features have a consistently higher chance of being heritable: they were 2.78x and 3.72x more likely to be heritable compared to the single tissue-based features and PCA based features.

The power of heritability test for PCA based cross-tissue TWAS is generally low, and the PC that captures the genetic signal best varies across scenarios (S3 Fig). The PCs that explain more of the variance in gene expression are not necessarily more heritable. Sometimes the second or third PC is heritable, but the first PC is not. We also observed that the chance of the PCA based feature to be heritable decreased as the correlation between the genetic effect of the correlated tissue and the causal tissue increased. This may occur because non-genetic sources of correlation in expression across tissues outweigh genetic sources when the genetic contributions to expression are highly correlated. In this setting, the top PCs often do not capture the genetic effects.

Because sCCA features are constructed by maximizing the correlation between gene expression and genotype, the Type I error rate for the cis-heritability test can be inflated due to overfitting. In fact, we did observe an inflated Type I error rate for heritability test under null for sCCA (S4 Fig). Considering individual features, the sCCA-feature1 had the highest Type I error rate at 0.43, while PC-feature1 had a slightly inflated type I error rate at 0.06 and the single tissue features maintained the Type I error rate at 0.05 level. But when we account for overall testing of 3 sCCA features, 3 PCS features and 22 single tissue features, the Type I error rate for at least one single tissue being significantly heritable at 0.05 level was 0.65 which is similar to the observed Type I error rate for at least one of the sCCA features being heritable. We note that standard TWAS pipelines typically do not adjust for the number of tissue features tested at the heritability stage. Most importantly, even though the cis-heritability test had an inflated rate of Type I error, the final Type I error rate for the sCCA-TWAS while testing for an association between predicted expression and phenotype was still well controlled (S5 Fig).

## sCCA-features increase power of cross-tissue TWAS

Next, we compare the power of various approaches to multi-tissue TWAS to detect gene-trait associations via simulation. We simulated genotype and expression data using linkage disequilibrium (LD) and expression correlation information from GTEx. We set the gene expression in one tissue to be causal for the phenotype and varied the variance explained by genotype for the causal tissue, number of tissues with gene expression correlated with the causal tissue and the corresponding correlation (see Methods for more details). All methods control the Type I error when expression is not associated with the outcome (S5 Fig). In simulations, we varied the correlation between the casual and correlated tissue, the proportion of other tissues correlated with the casual tissue, whether the test results from the causal tissue was observed or not, and the proportion of gene expression variation explained by genotype in the casual tissue (see Methods for details). In the simulation scenarios that we considered—all of which involved some correlation between the genetic contribution to gene expression in the causal tissue and at least one other tissue—we observed that the relative performance of different methods did not change as a function of the genetic correlation between the casual tissue and the correlated tissues, or the proportion of all tissues correlated with the casual tissue, or whether the causal tissue was analyzed (Fig 3, S1–S3 Tables).

We considered three sets of methods: (1) single tissue TWAS based approaches, which perform the single tissue based TWAS and either account for multiple testing using Bonferroni or GBJ corrections, or combine the test results using S-MultiXcan, or UTMOST [12,13]; (2) tests based on cross-tissue features (using PCA or sCCA to build cross-tissue features); and (3) combined test of both single-tissue based methods and cross-tissue feature based methods, using either Bonferroni or ACAT to adjust for multiple testing [15].

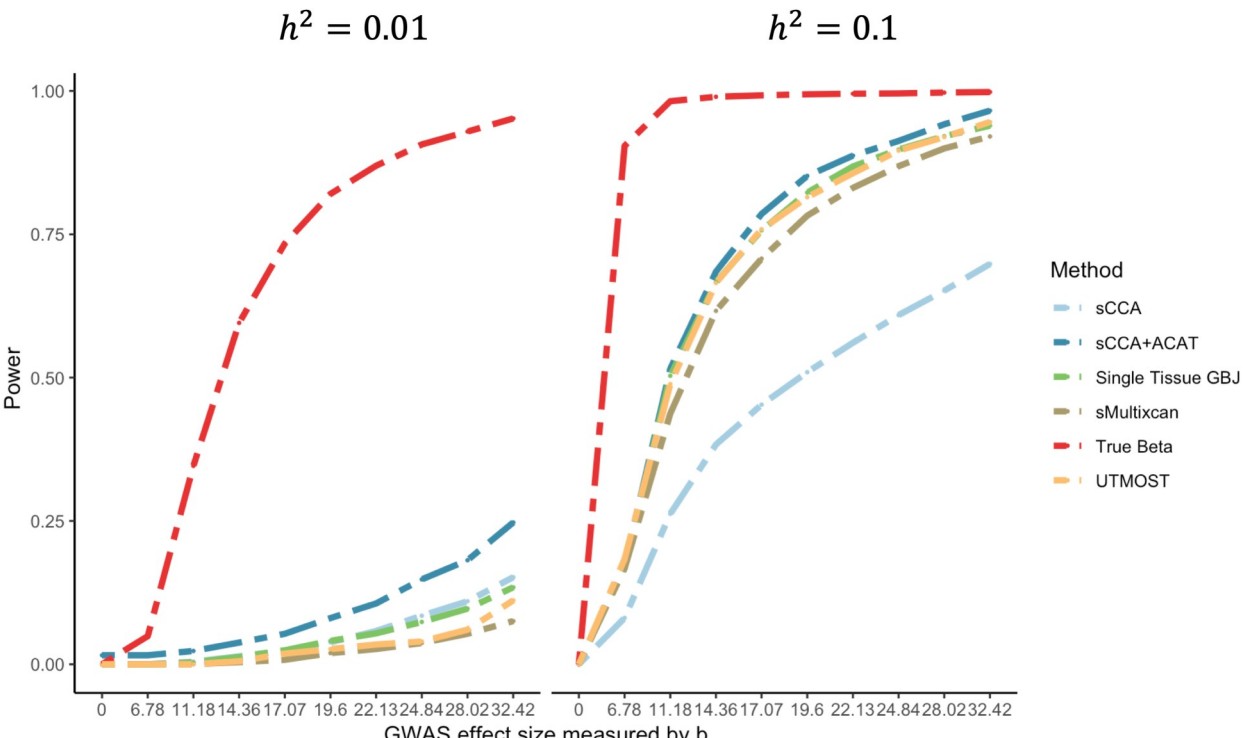

**Fig 3. Power comparison for cross-tissue TWAS methods.** Power (at $\alpha = 2.5 \times 10^{-5}$) as a function of GWAS effect size. For each tissue, we randomly sampled the z-scores from this multivariate normal and set b = $\sqrt{N_{gwas} \times r^2}$ to 0.00, 6.78, 11.18, 14.36, 17.07, 19.60, 22.13, 24.84, 28.02, 32.42 to achieve theoretical power of 5%, 10%,..., 90% at alpha level of 0.05. That is, when $r^2$ = 1% (when variation in gene expression in the target tissue explains 1% of the variability in the trait), the GWAS sample size $N_{gwas}$ ranges from 4,602 to 105,074. $h^2$ denotes the proportion of expression variance in the causal tissue explained by cis-genetic variation. sCCA+ACAT: combining 3 sCCA-features and 22 single-tissue tests with ACAT; sCCA: combining top 3 sCCA-features tests using a Bonferroni correction; Single Tissue_GBJ: combining 22 single-tissue TWAS statistics using the GBJ test; s-MultiXcan: combining 22 single tissue based test using s-MultiXcan); UTMOST: single tissue based approach with UTMOST; true weights: a TWAS test using the true (simulated) weights relating SNPs to expression in the causal tissue.

First, for the single tissue-based approaches, GBJ and S-MultiXcan had either similar power or GBJ had slightly higher power than S-MultiXcan (Fig 3, S1–S3 Tables). For example, when gene expression explains 2% of the variability in outcome and the GWAS sample size is 20,000, the average power of single-tissue test combined with GBJ and single-tissue combined with S-MultiXcan was 0.34, and 0.29, respectively. Second, for the approaches using cross-tissue features, sCCA yielded a substantially higher power than PCA under all scenarios (the average power is 0.26 for sCCA and $<10^{-4}$ for PCA; S1–S3 Tables). Third, for approaches to combine sCCA-TWAS and single tissue TWAS test results, combining sCCA-TWAS and single tissue TWAS test results with ACAT [15] yielded 1.37 times greater power than combining them with Bonferroni (the average power is 0.38 for ACAT and 0.37 for Bonferroni; S1–S3 Tables).

Finally, we compared single-tissue, cross-tissue, and combined single- and cross-tissue approaches. For simplicity, we only present comparisons between single-tissue based tests using GBJ to combine evidence across tissues, cross-tissue feature based approach with sCCA-features, and combined test of single-tissue based approach and sCCA-feature with ACAT, plus the recent S-MultiXcan and UTMOST approach [12,13].

Under the alternative, when gene expression has local genetic effects and gene expression is associated with the trait, the combined test of sCCA-features and single tissue-features using ACAT had the greatest power to detect a gene associated with the outcome, even when expression in the causal tissue was not directly measured (Fig 3). For example, when gene expression

explains 2% of the variability in outcome and the GWAS sample size is 20,000, the average power for the ACAT [15] combined test of sCCA features and single-tissue test, sCCA-TWAS and single-tissue tests combined with GBJ was 0.38, 0.23 and 0.34, respectively (S1–S3 Tables). The gain in power is likely because sCCA cross-tissue features are more likely to be significantly heritable, and thus increase the number of testable genes. This is particularly relevant for genes with low heritability: for such a gene, sCCA-TWAS has superior power (Fig 3 left panels). On the other hand, for highly heritability genes, single-tissue-based tests have better power than sCCA features. The combined test using both sCCA-TWAS and single-tissue TWAS results thus has superior power in both low- and high-heritability settings. Of note, the gain in power due to combining tests that perform well in different settings can be offset by the potential increased multiple testing burden. Fig 3 presents power comparisons under the scenario where half of the tissues are correlated with the causal tissue and the causal tissue is not observed (power under other scenarios are reported in S1–S3 Tables).

## sCCA-features provide insight into tissues where gene expression is associated with outcome

Although our primary motivation for combining multiple tissues when building expression weights is to increase the power of TWAS, since sCCA performs feature selection on the tissues as well as the cis-SNPs, it has the potential to suggest which tissues may be responsible for an identified TWAS association.

Fig 4 shows the sensitivity and specificity for the first sCCA component placing non-zero weight on the causal tissue (if included in the expression panel), or a tissue whose genetic

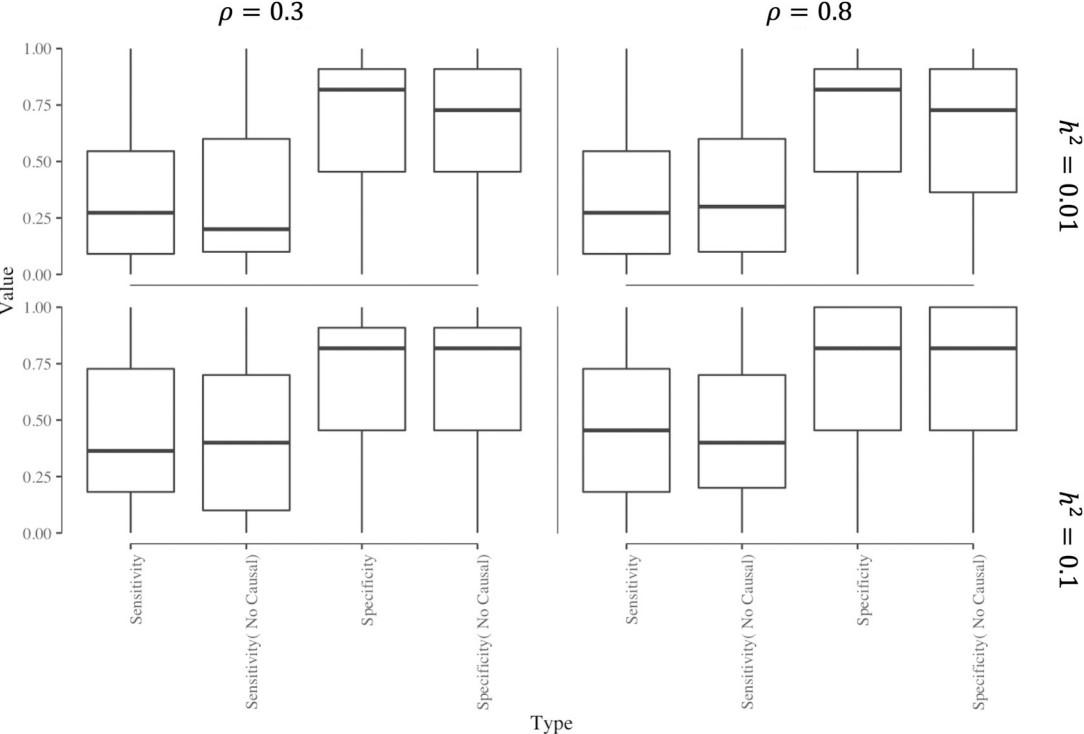

**Fig 4. Sensitivity and Specificity of sCCA features.** The box plot of sensitivity and specificity of sCCA putting non-zero weights on the tissue where genotype regulates gene expression. We varied underlying gene expression heritability ($h^2$) and correlation ($\rho$) with the causal tissue as: (a) $h^2 = 0.01$, $\rho = 0.3$; (b) $h^2 = 0.01$, $\rho = 0.8$; (c) $h^2 = 0.1$, $\rho = 0.3$; (d) $h^2 = 0.1$, $\rho = 0.8$.

contribution is correlated with that of the causal tissue. The sensitivity of the first sCCA component putting a non-zero weight on a causal or correlated tissue increases with the gene expression $h_g^2$ and the correlation between the causal tissue and the correlated tissues. Under our simulation assumption, the specificity of the first sCCA component is consistently high, which indicates that when combining gene expression across tissues with sCCA, it is less likely that non-relevant tissues would be included in the top sCCA expression feature. Thus, sCCA can effectively increase sample size while excluding noise. The tissues with non-zero weights in sCCA have a higher probability of being causal.

## sCCA performance is robust to missing data imputation in the expression data

The sCCA-TWAS approach requires a complete gene expression matrix: every individual used to train the sCCA features must have expression data from every tissue included in the analysis. However, this is typically not true for multi-tissue gene expression datasets like GTEx [9], where not all donors have samples or expression data from all tissues. A complete case analysis can greatly reduce the sample size available to train sCCA features. On the other hand, imputing missing expression data may induce measurement error or bias. We evaluate the impact of imputing missing expression data via simulation. We simulate complete gene expression and genotype data based on correlations in gene expression observed across GTEx; we then perform single-tissue based TWAS using weights trained in the complete data set. For sCCA and PCA based approaches, we mask the expression data matrix randomly based on the missing proportion pattern for each tissue in GTEx, then impute the missing expression data with MICE [18], using the "predictive mean matching" method. We then perform sCCA-TWAS or PCA-TWAS on the imputed gene expression dataset. sCCA-TWAS applied to imputed expression data still correctly controlled the Type I error rate. Although the power for sCCA-TWAS was lower when using imputed expression data (across all scenario decreased from 0.38 to 0.21), the sCCA-TWAS still provide valuable information when the genetic signal for gene expression is weak.

## Real-data application

**Applying sCCA to GTEx data increased the number of testable genes.** We applied the sCCA-TWAS approach based on top 3 sCCA-features to integrate GTEx data (version 6) and GWAS summary statistics data for 10 complex traits using the same *cis*-heritability filter as TWAS HUB, and compared the results with single tissue based TWAS results on TWAS HUB [17]. The phenotype information is included in Table 1 and the tissue expression dataset information is included in S4 Table. We choose to include top 3 sCCA-features as we observed in the simulation study that the gain in power due to including more features was negligible (S7 Fig). With sCCA cross-tissue features, we increased the number of testable genes to 21,740 compared to 12,027 (all GTEx tissues on TWAS HUB) and 18,954 (all panels on TWAS HUB). Among the genes that we could test using sCCA-TWAS, 10,649 genes were not testable in any of the other single-tissue panels available (that is, they did not pass the filtering criterion for cis-heritability or prediction strength set by TWAS-HUB). At the same time, with sCCA-features that combine expression profiles across multiple tissues, we reduced the multiple testing burden from 84,964 (GTEx tissues) and 157,316 (all panels in TWAS HUB) to 38,620. When the *cis*-genetic regulation is shared across multiple tissues, sCCA-TWAS reduces the redundancy in expression features tested. Using sCCA-TWAS as opposed to single-tissue TWAS increased the number of testable genes relative to single GTEx tissues by 81% and reduced the

**Table 1. Summary of data application results.**

| Trait | GWAS sample size | Number of significant loci | Number of significant genes in TWAS HUB using GTEx panel single-tissue weights | Number of significant genes by sCCA-TWAS | Number of significant genes by sCCA+ACAT |
|---|---|---|---|---|---|
| Alzheimer's Disease | 388,324 | 17 | 34 | 44 | 51 |
| Breast Cancer | 228,951 | 79 | 162 | 260 | 278 |
| Coronary Artery Disease | 56,422 | 11 | 11 | 8 | 11 |
| Type 2 Diabetes | 48,761 | 5 | 4 | 2 | 4 |
| Schizophrenia | 65,967 | 38 | 58 | 138 | 90 |
| BMI | 457,824 | 255 | 782 | 1132 | 1246 |
| Height | 458,303 | 423 | 2891 | 4080 | 5112 |
| Smoking Status | 457,683 | 59 | 106 | 166 | 164 |
| Chronotype | 410,520 | 69 | 82 | 145 | 140 |
| Tanning | 449,984 | 65 | 197 | 274 | 325 |

References: Alzheimer's disease [19], breast cancer [20], coronary heart disease [21], Type 2 Diabetes [22], Schizophrenia [23], Body mass index, height, smoking status, chronotype, and tanning [24].

multiple testing burden by 55%; realtive to all panels in TWAS HUB we increased the number of testable genes by 56% and reduced the multiple testing burden by 75% [17].

**Real-data application detects novel predicted-expression to phenotype associations.** The sCCA+ACAT and sCCA-feature TWAS detected additional associations between predicted gene expression and phenotype for the 10 GWAS traits we considered (Table 1). The single-tissue TWAS tests with GTEx weights identified 4,327 phenotype gene expression associations. In aggregate, sCCA-TWAS identified 4,400 additional associations for 10 phenotypes compared to single tissue GTEx TWAS, and the sCCA+ACAT combined test identified 3,277 additional associations compared to single tissue GTEx TWAS (Figs 5 and 6(A)). All the significant genes identified by sCCA+ACAT are reported in S5 Table. The two phenotypes with the largest number of associated genes identified are height and BMI, which are both highly polygenetic. To further contrast the significant associations identified, we considered the overlap between the associations identified with sCCA cross-tissue TWAS and single tissue TWAS for each phenotype. On an average, 18% of the gene-phenotype associations were identified by both single-tissue TWAS and sCCA TWAS, 49% gene-phenotype associations were only identified by sCCA-TWAS, and 34% signals were only detected by single tissue TWAS (Fig 6(B)).

ACAT served as a good combination method for single tissue and sCCA TWAS. Out of the total number of associations identified by either single-tissue TWAS, sCCA-TWAS, or sCCA +ACAT, 85% were significant in the sCCA+ACAT combined test. Among the gene-trait associations that were identified using the sCCA+ACAT approach, 41% were also identified by the single tissue approach but not the sCCA approach; 36% were also identified using the sCCA approach but not the single-tissue approach; 23% were identified using all three approaches; and 1% were identified using only the sCCA+ACAT combined approach. Fig 6(C) shows the breakdown in the testing performance by phenotype.

Direct comparison of the absolute z-scores from all the single tissue TWAS and sCCA-T-WAS shows a correlation of 0.86. The sCCA+TWAS absolute z-score is slightly greater than the median value of single tissue absolute z-score of the same gene from multiple tissues (S6 Fig).

We also used UTMOST to construct tissue-specific expression weights using all 44 GTEx tissues and conduct TWAS of the ten traits in Table 1. As expected, by leveraging similarities

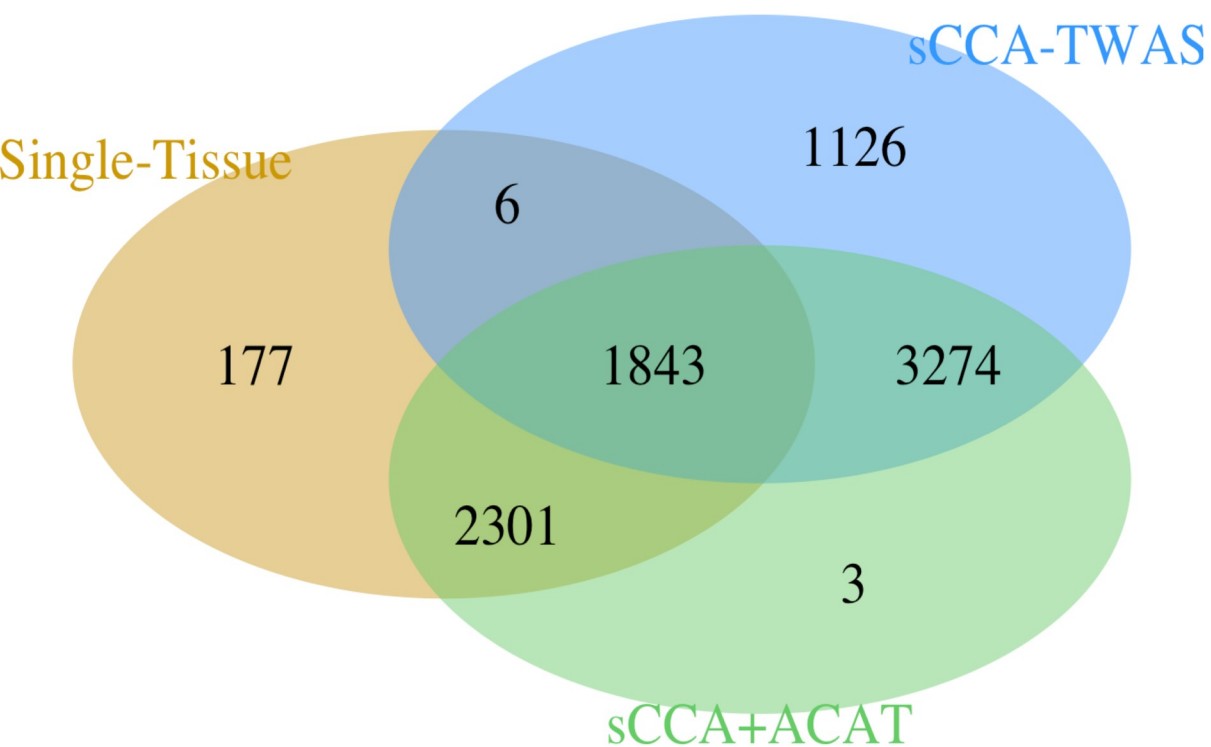

**Fig 5. Venn Diagram of the significant expression-phenotype associations.** The Venn Diagram of the significant expression-phenotype associations for single tissue test results, sCCA-TWAS test results and ACAT combined results (p<0.05 after accounting for testing multiple genes and multiple features). sCCA+ACAT: combining 3 sCCA-features and 22 single-tissue tests with ACAT; sCCA: combining top 3 sCCA-features tests using a Bonferroni correction; Single Tissue: combining 22 single-tissue TWAS statistics using Bonferroni.

in genetic regulation of expression across tissues, UTMOST detected more associations between predicted expression and phenotypes than single-tissue based models: 7,121 significant associations for UTMOST versus 4,327 for single-tissue GTEx TWAS (S6 Table). Moreover, although the sCCA+ACAT approach identified more associations (7,421) than UTMOST overall, UTMOST detected more significant associations for most traits (7/10; S6 Table). The relative performance of sCCA+ACAT and UTMOST in this particular application is due in part to the different expression training sets used: we applied UTMOST to data on up to 361 donors across 44 tissues and sCCA to data on 134 donors across 22 tissues.

## Discussion

We proposed a novel approach (sCCA-TWAS) to construct cross-tissue expression features using sparse canonical correlation analysis to boost the power of transcriptome-wide association studies. Through simulations we show that if the genetic component of gene expression in the causal tissue is correlated with the genetic contribution of expression in other tissues, then sCCA-TWAS has greater power than the approaches that use TWAS test statistics based on single-tissue features, including simply applying Bonferroni correction for the number of tissues tested or combining single-tissue tests using a GBJ procedure, S-MultiXcan, or UTMOST [12,13,16]. We have also proposed to combine sCCA-TWAS tests with single-tissue TWAS tests implementing the aggregate Cauchy association test (sCCA+ACAT). sCCA+ACAT achieves optimal or near-optimal power among the procedures considered both when the causal tissue is genetically correlated with other tissues and when it is not, suggesting that the sCCA+ACAT is a useful method when the genetic architecture of tissue-specific expression

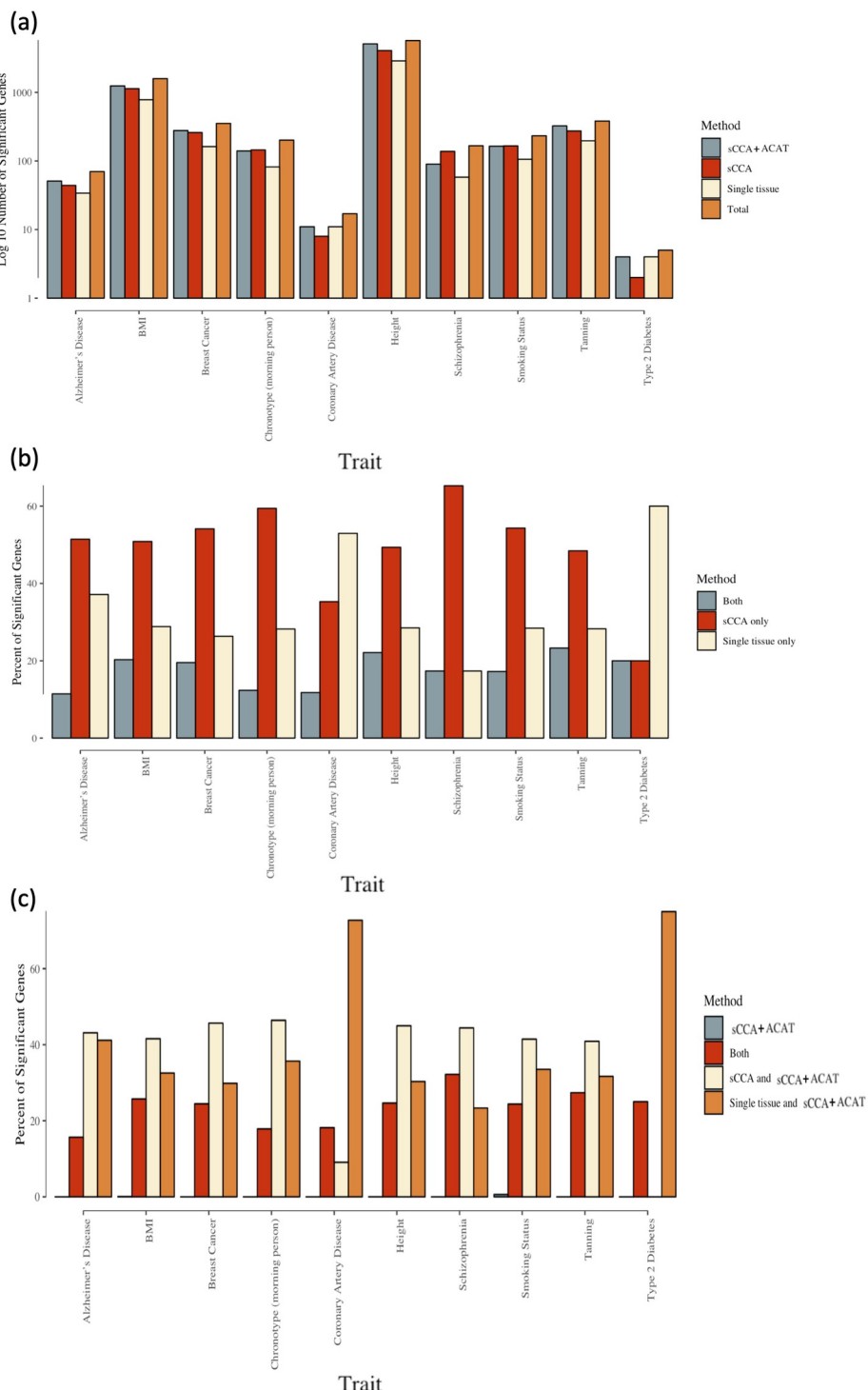

**Fig 6. (a) Number of significant genes identified by ACAT combined test, sCCA-TWAS, TWAS using single tissue GTEx data and the total number of significant genes identified by all three methods.** Different phenotypes are arranged along the x-axis and the number of significant genes identified by ACAT combined test, sCCA+TWAS, TWAS using single-tissue GTEx data and the total number of significant genes identified by all three methods are shown in the y-axis on log 10 scale. The information about the phenotype are provided in Table 1. sCCA+ACAT: combining 3 sCCA-features and 22 single-tissue tests with ACAT; sCCA: combining top 3 sCCA-features tests using a Bonferroni correction; Single Tissue: combining 22 single-tissue TWAS statistics using Bonferroni. **(b) Percentage of significant associations identified by both single tissue TWAS and sCCA TWAS, by only sCCA-TWAS, and by**

**only identified by single tissue TWAS, among all associations identified with sCCA cross-tissue TWAS or single tissue TWAS.** Different phenotypes are arranged along the x-axis and the percentage of significant identified by both single tissue TWAS and sCCA-TWAS, by only sCCA-TWAS, and by only identified by single tissue TWAS are shown in the y-axis. The information about the phenotype are provided in Table 1. **(c) Percent of significant identified by only sCCA+ACAT, by sCCA+ACAT, sCCA-TWAS and single tissue TWAS, by both sCCA-TWAS and sCCA +ACAT, by both single tissue TWAS and sCCA+ACAT among all significant genes.** Different phenotypes are arranged along the x-axis and the percentage of significant associations by only ACAT, by ACAT, sCCA-TWAS and single tissue TWAS, by both sCCA-TWAS and ACAT, by both single tissue TWAS and ACAT are shown in the y-axis. The information about the phenotype are provided in Table 1. sCCA+ACAT: combining 3 sCCA-features and 22 single-tissue tests with ACAT; sCCA: combining top 3 sCCA-features tests using a Bonferroni correction; Single Tissue: combining 22 single-tissue TWAS statistics using Bonferroni.

and its relationship to outcome is unknown. This increase in power is due in part to the greater number of genes with significantly heritable sCCA features relative to single-tissue features. sCCA-TWAS also greatly improved power relative to another cross-tissue technique using PCA to create cross-tissue features, as the leading principal components often capture non-genetic sources of covariation in gene expression (a general drawback to cross-trait association analysis using PCA[25]). Moreover, the tissue-wise loadings from sCCA factors associated with outcome may provide some guidance to which tissues are causally related to the outcome (or genetically correlated with the unmeasured causal tissue).

When developing expression feature weights for inclusion in TWAS, there is a trade-off between Type I and Type II errors at the expression-modeling stage. Stringent control of the Type I error when testing expression-feature heritability will increase Type II error and reduce the number of genes that can be tested at the TWAS stage. S4 Fig highlights that fact that the sCCA procedure identifies more features for testing at the TWAS testing stage, in part because it is implicitly making a different choice regarding the trade-off between Type I and Type II error at the expression-modeling stage. Importantly, we show that when the TWAS weights for the sCCA features are not associated with expression (Type I errors at the expression-modeling stage), Type I error is controlled at the TWAS testing stage, so the final TWAS associations retain appropriate control of false positives. We also show that including sCCA feature weights can improve power, even if some of the tested weights are not associated with expression. This suggests that on balance the additional genes that can be tested using sCCA feature weights include enough genes with truly genetically associated expression levels to more than counter balance the penalty from testing features that are not truly associated with genetic variation.

sCCA- and sCCA+ACAT- TWAS can be useful in a situation where eQTL data on germline genetic variation and expression in multiple tissues or cell-types are available on the same set of individuals. sCCA-TWAS cannot be directly applied when eQTL data on different tissues are available on different, non-overlapping samples. When both a multi-tissue reference panel (such as GTEx) and additional large single-tissue reference panels are available, sCCA+ACAT can make use of both the cross-tissue features from the multi-tissue panel and the independent single-tissue panels. Finally, inferring the causal tissue from a set of cross-tissue or single-tissue TWAS results remains an important open question. Although the tissue weights from the sCCA features may provide some clues, further work is needed to develop principled sensitive and specific methods for identifying candidate causal tissues.

Our primary goal was to improve TWAS power by constructing TWAS weights that leverage pleiotropic effects on expression in multiple tissues. Using sCCA to construct cross-tissue features provides one promising approach, but other approaches may also increase power or better identify likely causal tissues. Moreover, missing data in multi-tissue expression panels—for example, some donors may be missing expression values for some tissues—can present

practical challenges. UTMOST uses a principled penalized regression approach to address incomplete data while leveraging potential similarities in genetic regulation of gene expression across tissues, but builds expression weights on a tissue-by-tissue basis. We used a generic method to impute missing gene-expression data (MICE) when constructing sCCA features and corresponding weights; other techniques tailored to multi-tissue gene expression data may further improve the efficiency of sCCA features. Another multi-tissue TWAS method, MR-TJI [26], that combines Mendelian randomization and joint tissue imputation techniques, was published during the review process of this work. The MR-TJI method jointly models the cross-tissue gene expression to develop tissue-specific genetic predictors of expression and then performs tissue-specific TWAS tests, similar to UTMOST[12]. We were unaware of this approach before it was published while we were revising this paper, so we could not include a direct comparison of our proposed methods and the MR-TJI approach.

## Methods

### sCCA

Suppose that we have n observations on $p_1+p_2$ variables, and the variables are naturally partitioned into two groups of $p_1$ and $p_2$ variables, respectively, where $p_1$ is the number of variables in the first group and $p_2$ is the number of variables in the second group. Let $\mathbf{G} \in R(n \times p_1)$ correspond to the first set of variables, and let $\mathbf{X} \in R(n \times p_2)$ correspond to the second set of variables. Assume that the columns of $\mathbf{G}$ and $\mathbf{X}$ have been standardized to have mean zero and standard deviation one. In our setting, $\mathbf{G}$ is a matrix of standardized genotypes with SNPs corresponding to the columns and $\mathbf{X}$ is a matrix of tissue-specific gene expression values with tissues corresponding to the columns.

Standard CCA seeks canonical vectors $\boldsymbol{\omega_1} \in R(p_1)$ and $\boldsymbol{\omega_2} \in R(p_2)$ that maximize correlation between $\mathbf{G}\boldsymbol{\omega_1}$ and $\mathbf{X}\boldsymbol{\omega_2}$ [14], that is:

$$\text{maximize}_{\omega_1,\omega_2} \ \boldsymbol{\omega}_1^{T} \boldsymbol{G}^T \mathbf{X} \boldsymbol{\omega_2} \text{ subject to } \boldsymbol{\omega}_1^T \boldsymbol{G}^T \boldsymbol{G} \boldsymbol{\omega_1} = \boldsymbol{\omega}_2^T \boldsymbol{G} \boldsymbol{\omega_1} = \boldsymbol{\omega}_2^T \boldsymbol{X}^T \boldsymbol{X} \boldsymbol{\omega_2} = \mathbf{1}$$

However, CCA is not appropriate when $p_1, p_2 \approx n$ or $p_1, p_2 >> n$. Witten et al. [14] proposed sparse CCA, a penalized version of CCA, by adding $L_1$ and $L_2$ penalization in the previous optimization problem [14] as:

$$\text{maximize}_{\omega_1,\omega_2} \ \boldsymbol{\omega}_1^{T} \boldsymbol{G}^T \mathbf{X} \boldsymbol{\omega_2} \text{ subject to } \boldsymbol{\omega}_1^T \boldsymbol{G}^T \boldsymbol{G} \boldsymbol{\omega_1} \leq 1, \boldsymbol{\omega}_2^T \boldsymbol{X}^T \boldsymbol{X} \boldsymbol{\omega_2} \leq 1,$$

and $||\boldsymbol{\omega_1}||_1 \leq c_1, ||\boldsymbol{\omega_2}||_1 \leq c_2$.

Using the identity matrix $\mathbf{I}$ as a substitute for $X^T X$ and $G^T G$ gives what can be termed as "diagonal penalized CCA", and the optimization problem can be re-formulated as:

$$\text{maximize}_{\omega_1,\omega_2} \ \boldsymbol{\omega}_1^{T} \boldsymbol{G}^T \mathbf{X} \boldsymbol{\omega_2} \text{ subject to } ||\boldsymbol{\omega_1}||_2^2 \leq 1, ||\boldsymbol{\omega_2}||_2^2 \leq 1, ||\boldsymbol{\omega_1}||_1 \leq c_1, ||\boldsymbol{\omega_2}||_1 \leq c_2$$

For a small $c_1$ and $c_2$, this results in $\boldsymbol{\omega_1}$ and $\boldsymbol{\omega_2}$ to be sparse, i.e., many of the elements of $\boldsymbol{\omega_1}$ and $\boldsymbol{\omega_2}$ will be exactly equal to zero. Witten et al. proposed to solve this maximization problem by choosing an initial $\boldsymbol{\omega_2}$ s.t. $||\boldsymbol{\omega_2}||_2 = 1$, and then iteratively maximizing $\boldsymbol{\omega}_1^{T} \boldsymbol{G}^T \mathbf{X} \boldsymbol{\omega_2}$ subject to $L_1$ and $L_2$ constraints for $\boldsymbol{\omega_1}$ and $\boldsymbol{\omega_2}$ in turn [14]. The sparsity parameters $c_1$ and $c_2$ are chosen using a permutation procedure, where rows in $\mathbf{X}$ are randomly permuted $k$ times. sCCA is performed on the original data set and each of the permuted data sets across an ordered set of $c_1$ and $c_2$ pairs chosen to represent increasing penalization. The pair that maximizes the absolute difference between the Fisher-transformed correlation cor$(\mathbf{G}\boldsymbol{\omega_1}, \mathbf{X}\boldsymbol{\omega_2})$ based on sCCA applied to the original data set and average the Fisher-transformed cor$(\mathbf{G}\boldsymbol{\omega}_1^*, \mathbf{X}^* \boldsymbol{\omega}_2^*)$ based on sCCA applied to the permuted data $\mathbf{X}^*$. We set $k = 15$. Given this pair of sparsity parameters, we

generate subsequent canonical variates by repeatedly applying the sCCA algorithm to the new correlation matrix $G^T X$ after regressing out the previous canonical components [14].

## TWAS

The TWAS pipeline consists of three steps: first, identifying gene expression features that have positive cis-heritability; second, building a linear predictor for each cis-heritable gene feature; and third, constructing the TWAS test statistic combining the prediction weights and summary Z-scores from a trait GWAS.

We computed the p-values for testing $cis\text{-}h_g^2 = 0$ using a likelihood ratio test implemented in GCTA that compares a model with a local random genetic effect to a model without a genetic effect [27]. We included all SNPs that fall within 500 kb of the transcription start and stop sites of a gene. We removed the genes that failed the heritability test from the set of candidate genes, and only the genes with a significant heritability were included in the subsequent prediction model construction.

We then used Elastic Net penalized regression implemented in the R package glmnet [28] to construct linear genetic predictors of gene expression features $W$ based on all the *cis* SNPs in the eQTL reference panel (500 base-pair window surrounding the transcription start and stop sites). We use Elastic Net applied to all cis SNPs rather than restricting to the SNPs selected in the sCCA procedure because Elastic Net has better performance predicting expression features than $L_1$-penalized regression [3,4]. We applied 5-fold cross-validation to choose the elastic net penalty parameters.

We calculated the TWAS test statistic as $Z_{TWAS} = wZ/(w\Sigma_{s,s}w')^{1/2}$, where $Z$ is a vector of standardized effect sizes of SNPs for a trait in the *cis* region of a given gene (Wald z-scores), and $w = (w_1\ w_2\ w_3.\dots.w_j)$ is a vector of prediction weights for the expression feature of the gene being tested, and $\Sigma_{s,s}$ is the LD matrix of the cis SNPs estimated from the 1000 Genomes Project as the LD reference panel. Under null hypothesis that there is no association between the gene expression feature and phenotype, $Z_{TWAS}$ should follow a normal distribution with mean zero and variance one.

## sCCA-TWAS

Consider a gene expression array of a certain gene for *n* individuals and $p_2$ tissues $X_{nxp_2}$, and the genotype data $G_{nxp_1}$ for the same set of individuals at $p_1$ cis-SNPs. Assume that the columns of $X_{nxp_2}$, and $G_{nxp_1}$ have been standardized to have mean zero and variance one.

We apply sCCA (described above) and extract the first three pairs of canonical vectors: $(\boldsymbol{\omega}_1^{(1)}, \boldsymbol{\omega}_2^{(1)})$, $(\boldsymbol{\omega}_1^{(2)}, \boldsymbol{\omega}_2^{(2)})$ and $(\boldsymbol{\omega}_1^{(3)}, \boldsymbol{\omega}_2^{(3)})$. We define three sCCA features as $X\boldsymbol{\omega}_2^{(1)}$, $X\boldsymbol{\omega}_2^{(1)}$ and $X\boldsymbol{\omega}_2^{(3)}$. Then we treat the three sCCA-features as three repeated measure of gene expression across tissue and apply TWAS procedure to them, record the p-value for heritability and z-score of these three features. We account for testing multiple sCCA features per gene via Bonferroni's correction, including only the tests where the sCCA-feature passed the heritability test. We decided to include at most 3 sCCA features, because in simulations, the power gain from including more features appears to be small (S7 Fig).

## Single tissue test based cross-tissue TWAS

As a comparison, we also considered single-tissue test based cross-tissue TWAS, where we perform TWAS on the gene expression in each tissue, record the z-scores and p-values for heritability test, respectively. We account for testing multiple tissues for each gene via i) a Bonferroni multiple testing correction or ii) a generalized Berk-Jones (GBJ) test with single-

tissue association statistics $Z$ and their covariance matrix $\Sigma$ as inputs [16]. We estimate $\Sigma$ as $W\Sigma_{s,s}W'$, where $W_{qxp}$ is a matrix with the expression weights for each tissue in each row and each SNP [12].

## Combined test with sCCA-features and single-tissue features

While sCCA can increase power when sample sizes in individual tissues are small and the genetic contribution to expression is shared across tissues, a single-tissue based approach may be more powerful when the genetic contribution to expression in the causal tissue is uncorrelated with genetic contribution to expression in other tissues. Thus, a combined test for sCCA-features and single-tissue features can have a better average power across a range of scenarios. We therefore consider approaches that combine sCCA and single-tissue expression features, accounting for testing multiple features per gene using a Bonferroni correction, the GBJ test [16], or the ACAT [15]. The GBJ test is a set-based test proposed for GWAS setting, which extended the Berk-Jones (BJ) statistics by accounting for correlation among tests [16]. ACAT is a fast p-value combination method that uses Cauchy distribution to approximate the distribution of a weighted sum of transformed p-values. ACAT has been shown to work well in the context of genetics research, mainly because it does not require the estimation of correlation structure among the combined p-values.

## PCA based cross-tissue TWAS

We also considered aggregating across tissue signal through Principal Component Analysis (PCA). We first applied PCA to the gene expression matrix $\mathbf{X_{nxq}}$, then used the top 3 principal Components (PCs) as new feature for TWAS. We accounted for testing multiple PCs for each gene by Bonferroni adjustment, including only the tests where the PCs passed the heritability test.

## S-MultiXcan

Summary-MultiXcan (S-MultiXcan) is another single-tissue based approach for generating multi-tissue gene expression, and draw phenotype associations inference. It utilizes the LD information from a reference panel to integrate univariate S-PrediXcan results. It consists of the following steps: (1) computation of single tissue association test statistics $\hat{Z}$ with S-PrediXcan [2]; (2) estimation of the correlation in tissue-specific predicted gene expression levels using the LD information from a reference panel (typically GTEx or 1000 Genomes); (3) discarding components of smallest variation from the matrix of correlations in genetically-predicted tissue-specific gene expression levels to avert collinearity and numerical problems (singular value decomposition, analogous to PC analysis in individual-level data). (4) estimation of multi-tissue test statistics from the univariate (single-tissue) results with the help of expression correlation.

The aggregate S-MultiXcan test statistic is then calculated as $\hat{Z}^T Cor(X)^+ \hat{Z} \sim \chi^2_k$, where $Cor(X)^+$ is the pseudo-inverse of a SVD-regularized version of the correlation matrix of $X$, and k the number of components surviving the SVD pseudo-inverse (the regularized version of the correlation matrix is formed by decomposing the correlation matrix into its principal components and removing those eigenvectors corresponding to the eigenvalues $\frac{\lambda_{max}}{\lambda_i} < 30$).

## UTMOST

UTMOST [12] is also a single-tissue based approach for testing multi-tissue gene expression and phenotype associations. To construct tissue-specific TWAS weights, it jointly models the

relationship between gene expression levels across multiple tissues and genotypes using grouped panelized regression. Then, it tests the associations between the trait and gene expression in each tissue. Lastly, combines the single-tissue test results with GBJ.

## Data simulation settings

We simulated genotype and expression data using linkage disequilibrium (LD) and expression correlation information from the Genotype-Tissue Expression project (GTEx) version 6 [9]. GTEx includes data from 449 donors across 44 tissues, with tissue-specific sample sizes ranging from 70 to 361. We removed: (1) individuals with data available for less than 40% of the tissues and (2) tissues where less than 30% of the individuals have data. This results in a 134 ($n$ ndividuals) by 22 ($p_2$ tissues) ordered expression matrix for each gene. We randomly sampled 400 genes in the data set, extracted the cis-SNPs within 500kb around the gene boundary (number of cis-SNPs indicated by $p_1$) and the gene expression for these 134 individuals and 22 tissues, and imputed missing expression values with the column mean. We used this data set to calculate the correlation among gene expression levels across tissues ($\Sigma_X$) and the LD structure of cis-SNPs ($\Sigma_G$) for each of the 400 genes.

Individual-level data for a gene expression reference panel data were generated assuming that the gene expression for a particular gene in tissue $i$ is $\mathbf{X_i} = \mathbf{G}\boldsymbol{\beta_i} + \epsilon_i$, where $\mathbf{G}$ is the local genotype matrix, $\boldsymbol{\beta_i}$ is the weight for genotype on gene expression in tissue $i$, and the residuals $\epsilon_i$ are normally distributed, independent across individual but correlated across tissues. We generated each row in the $n \times p_1$ genotype matrix $\mathbf{G}$ as $MVNp_1$ with mean zero and variance-covariance matrix $\Sigma_G$, the LD matrix calculated from the GTEx genotype data from the gene's cis region. We randomly sampled one tissue to be causal and $N_{corr}$ tissues to be genetically correlated with the causal tissue. We selected 3% of the cis-SNPs to be causally related to gene expression in the causal tissue and sampled their weights for gene expression, $\beta_{ij}^{causal}$ from normal distribution with mean zero and variance $h_g^2$; the remaining $\beta_{ij}^{causal}$ for $j$ not in the set of causal SNPs were set to 0. To reflect the genetic correlation $\rho$ between the causal tissue and the $N_{corr}$ genetically correlated tissues, the weight for the same SNPs in the correlated tissues were sampled as

$$\boldsymbol{\beta_{\mathbf{correlated}}} \sim \mathrm{MVN}_{\mathrm{Ncorr}} \times 1(\boldsymbol{\beta_{\mathbf{causal}}} \times \rho \times \mathbf{1_{Ncorr}}, (1 - \rho^2) \times h_g^2 \cdot \mathbf{I_{Ncorr}})$$

This resulted in a $p_1$ by $p_2$ weight matrix for genotype on tissue-specific gene. Residual gene expression values were simulated as:

$$\mathbf{e} \sim \mathrm{MVN}_{\mathrm{n \times p}}(\mathbf{0}, \mathbf{diag}(\boldsymbol{s_e}) \times \boldsymbol{\Sigma_X} \times \mathbf{diag}(\boldsymbol{s_e}))$$

where $s_e = \sqrt{\mathrm{Var}\left(\mathbf{X}\boldsymbol{\beta}_{\mathrm{q \times p}}\right) \times \left(\frac{1}{h_g^2} - 1\right)}$, so that the variance in gene expression explained by genotype in each tissue is $h_g^2$. We considered four scenarios, defined by combination of the proportion of tissues genetically correlated with the causal tissue and whether the causal tissue was observed in the analysis: all or half of the tissues were correlated with causal tissue; the causal tissue was or was not observed. We varied $h_g^2$ from 0.01 to 0.1, and the genetic correlation coefficient $\rho$ between the causal and other tissues from 0.3 and 1.

Given the SNP-expression weights in a tissue and assuming that the trait under study Y has unit variance and the true mean of the trait is related to expression levels in the causal tissue via $E[Y] = r\, X_{causal}$, the cis-SNP GWAS z-scores for tissue $i$ are distributed as $\mathbf{Z} \sim \mathrm{MVN}$ $(\Sigma_G \times b \times \boldsymbol{\beta i}, \Sigma_G)$, where $b = \sqrt{N_{gwas} \times r^2}$. For each tissue, we randomly sampled the z-scores from this multivariate normal and set b to 0.00, 6.78, 11.18, 14.36, 17.07, 19.60, 22.13, 24.84,

28.02, 32.42 to achieve the theoretical power of 5%, 10%,. . ., 90% at alpha level of 0.05. For example, when $r^2$ equals 1% (i.e., variation in gene expression in the target tissue explains 1% of the variability in the trait), the GWAS sample size $N_{gwas}$ ranges from 4,602 to 105,074. We repeated the whole procedure on 400 randomly selected genes. For each gene, we further replicated 5 times for a total of 2000 replicates. For each statistical test procedure (sCCA, PCA, s-MultiXcan, etc.), and for each replicate, there are three possible outcomes: A: the gene is not heritable [i.e., no sCCA feature is significantly heritable, or no PCA, or no single tissue, depending on the procedure]; B: the gene is heritable but not significantly associated with the trait (after accounting for multiple testing across heritable tissues/features); and C: the gene is heritable and significant. We calculate Type I error as C/(B+C) and power as C/2000. The significance threshold used for Type I error calculation was 0.05. For power calculations, the significance threshold was $2.5 \times 10^{-5}$; this threshold approximates a Bonferroni correction for the number genes with at least one heritable feature (i.e. the number of genes tested in a TWAS).

## Data application

We applied sCCA-TWS approach to GTEx and 10 real life-style, polygenic complex traits and diseases (Table 1): Alzheimer's disease [19], breast cancer [20], coronary heart disease [21], Type 2 Diabetes [22], Schizophrenia [23], Body mass index, height, smoking status, chronotype, and tanning [24]. Before applying sCCA to the GTEx data (version 6), we removed individuals with data available in less than 40% of the tissues. We also removed tissues where less than 30% of the donors have sample. This resulted in a 134 (n) individual by 22 ($p_2$) tissue expression matrix for each gene (list of tissues provided in S4 Table). We imputed the missing expression data using the predictive mean method in R package MICE [18]. We performed sCCA on the imputed gene expression and genotype data from GTEx, extracted the top 3 canonical vectors for gene expression for each gene, and built three sCCA-features for each of the gene. Then we adopted the standard TWAS pipeline with the sCCA features, filtering out sCCA-features that failed to converge in GCTA or had a heritability test p-value greater than 0.01. We built linear genetic weights with the remaining sCCA-features using Lasso, Elastic Net (eNet), and top eQTL models, and performed TWAS with the model of highest cross validation $R^2$.

## Supporting information

**S1 Fig. Proportion of significant (p<0.05) heritability tests for different expression features when cis genetic variation is associated with expression in *all* tissues.** $\rho$ denotes the strength of the genetic correlation between expression in the causal tissue and tissues where expression is also associated with cis germline variation ("correlated tissues"). "Non-correlated tissues" are tissues where local germline variation is not associated with gene expression. Here expression in all of the tissues is genetically correlated with the causal tissue, and the causal tissue is not observed (performance in the causal tissue is included as a reference). PC1 is the first principal component of cross-tissue gene expression; sCCA-feature1 is the linear combination of tissue expression values from the first pair of sCCA canonical variables. $h^2$ denotes the proportion of expression variance in the causal tissue explained by cis genetic variation. (TIFF)

**S2 Fig. Proportion of significant (p<0.05) heritability tests for different expression features when cis genetic variation is associated with expression in *some* tissues.** $\rho$ denotes the strength of the genetic correlation between expression in the causal tissue and tissues where expression is also associated with cis germline variation ("correlated tissues"). "Non-correlated

tissues" are tissues where local germline variation is not associated with gene expression. Here expression in half of the tissues is genetically correlated with the causal tissue, and the causal tissue *is* observed. PC1 is the first principal component of cross-tissue gene expression; sCCA-feature1 is the linear combination of tissue expression values from the first pair of sCCA canonical variables. $h^2$ denotes the proportion of expression variance in the causal tissue explained by cis genetic variation.
(TIFF)

**S3 Fig. Proportion of significant (p<0.05) heritability tests for the top three principal components summarizing gene expression across features (*half* of the tissues are correlated with the causal tissue and causal tissue *not* observed).** $\rho$ denotes the strength of the genetic correlation between expression in the causal tissue and tissues where expression is also associated with cis germline variation. Half of the tissues are genetically correlated with the causal tissue, which is not observed. $h^2$ denotes the proportion of expression variance in the causal tissue explained by cis genetic variation.
(TIFF)

**S4 Fig. Type I error rate for cis-heritability tests.** Proportion of simulations where local genetic variation was nominally statistically significantly associated with gene expression, in the scenario where no association was present. sCCA-Feature_1: testing only the leading sCCA expression feature at the $\alpha = 0.05$ level; PCA-feature_1: testing only the lead cross-tissue expression principal component at the $\alpha = 0.05$ level; All_PCA-features and All_sCCA-features: proportion of simulations where at least one of the top three PCA (resp. sCCA) features was significant at the $\alpha = 0.05$ level; All_single_tissue: proportion of simulations where at least one of the 22 single-tissue tests was significant at the $\alpha = 0.05$ level.
(TIFF)

**S5 Fig. Type I error rate for cross-tissue TWAS methods.** Proportion of significant results under the (gene expression not associated with phenotype) averaged over all scenarios.
(TIFF)

**S6 Fig. Comparison of the absolute z-score for sCCA-TWAS and single tissue TWAS using weights calculated form GTEx data and GWAS summary statistics from 10 complex traits.** The TWAS test statistics using sCCA feature 1 and all single tissue weights from Fusion are plotted on the x-axis and y-axis respectively. The blue line is the fitted regression line and red line is y = x.
(TIFF)

**S7 Fig. Cumulative power for identify heritable gene when include sCCA feature 1 to feature 3.** The Y axis indicate the cumulative power of detecting heritable genes when include only sCCA feature 1, sCCA feature 1 and 2, and sCCA feature 1 to 3, average over all scenarios.
(TIFF)

**S1 Table. Summary of simulation power when gene expression in other tissues *not* correlated with the causal tissue.**
(XLSX)

**S2 Table. Summary of simulation power when gene expression in *half* of the tissues correlated with the causal tissue.**
(XLSX)

**S3 Table. Summary of simulation power when gene expression in *all* of the tissues correlated with the causal tissue.**
(XLSX)

**S4 Table. Summary of GTEx expression data.**
(XLSX)

**S5 Table. All significant genes identified by sCCA+ACAT for 10 selected phenotypes.**
(XLSX)

**S6 Table. Data Application Results for UTMOST.**
(XLSX)

## Author Contributions

**Conceptualization:** Bogdan Pasaniuc, Peter Kraft.

**Formal analysis:** Helian Feng.

**Investigation:** Bogdan Pasaniuc, Peter Kraft.

**Methodology:** Helian Feng.

**Visualization:** Helian Feng.

**Writing – original draft:** Helian Feng.

**Writing – review & editing:** Nicholas Mancuso, Alexander Gusev, Arunabha Majumdar, Megan Major, Bogdan Pasaniuc, Peter Kraft.

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
