## [Decision Letter · Decision Letter 0]

19 Aug 2020

Dear Dr Kraft,

Thank you very much for submitting your Research Article entitled 'Leveraging expression from multiple tissues using sparse canonical correlation analysis (sCCA) and aggregate tests improves the power of transcriptome-wide association studies (TWAS)' to PLOS Genetics. Your manuscript was fully evaluated at the editorial level and by independent peer reviewers. The reviewers appreciated the attention to an important problem, but raised some substantial concerns about the current manuscript. Based on the reviews, we will not be able to accept this version of the manuscript, but we would be willing to review again a much-revised version. We cannot, of course, promise publication at that time.

If you decide to revise the manuscript for further consideration at PLOS Genetics, please aim to resubmit within the next 60 days, unless it will take extra time to address the concerns of the reviewers, in which case we would appreciate an expected resubmission date by email to plosgenetics@plos.org.

[LINK]

We are sorry that we cannot be more positive about your manuscript at this stage. Please do not hesitate to contact us if you have any concerns or questions.

Yours sincerely,

Michael P. Epstein

Associate Editor

PLOS Genetics

David Balding

Section Editor: Methods

PLOS Genetics

Reviewer's Responses to Questions

**Comments to the Authors:**

Reviewer #1: The paper describes a generalization of the TWAS pipeline that uses data across multiple tissue. The generalization has the potential for increasing power of detecting associations. It seems to me that the main contribution of the paper is the use of multi-tissues gene expression features (weighted combination of gene expression measurements) in the TWAS pipeline. The idea of using multiple tissues is not new (see for example MultiXcan) but the proposed implementation here is novel - it relies on constructing better features using sparse canonical correlation. The performance of the method in simulations and real data is investigated. The Abstract describes a models increase in power when compared to other methods.

Is there software available for this methodology? How about a database of features? I am not sure how the community will benefit from this paper without them.

The sCCA-TWAS methodology as described is insufficient. As the authors mention in the paper in the context of GTeX data - not all subjects have measurements for all tissues. This is an interesting methodological problem that is also essential for an efficient implementation - how to deal with missing gene expression data? There are challenges due to the non-random nature of missing data (for example, when one brain tissues is missing, it is very likely that most other brain tissues will not be available as well). The simple solution of using only the subset of subjects that have data for all tissues is clearly inefficient. The proposed solution (on page 14) is also unsatisfactory - the authors propose to use an off-the-shelf method for imputation, MICE, that has not been validated as an accurate method for gene expression imputation. The proposed method controls Type 1 error (it is trivial to see why) but loses a great amount of power compared to complete data (0.38 to 0.21). It is clear that the authors miss an opportunity to increase power by doing a careful evaluation of how to deal with missing expression data. This can be done in the context of sparse canonical correlation via structured equations or by using imputation methods that have been developed in the context of gene expression data.

The sCCA methodology relies on the selection of tuning parameters that establish the level of sparsity in the genotype and expression data. There is a brief mentioning of using cross-validation for choosing c1 and c2 - but there are no details on how many folds should one select as a function of sample size and whether missing tissues have any impact on that choice. Note that cross-validation works poorly in regions with a lot of LD and when the number of subjects is small.

A clarification is needed for the weight building step after sCCA: are the regressions run on all cis SNPs or only on the SNPs that have non-zero weights in the sCCA step? It seems from the paper that it is all SNPs - but it is not clear on why it is better to ignore the information (derived weights) from the sCCA step.

A couple of naive ways to do the multi-tissue analysis are: (i) average the expression across tissues (equal weights, partly justified by the cross-tissue findings for eQTls); (ii) weights that maximize the correlation with the top cis-eQTL (as opposed to using all SNPs). Does the sCCA approach work better than these naive approaches?

On Table 1 - it is not clear if sCCA outperforms the competitors. I don't see it - but maybe I am reading the Table data incorrectly. The description on page 19 is not reflected in Table 1 - some clarification is needed.

Minor comments:

(i) someone needs to look at the references; for example 9 (author?) and 16 (published in 2019).

(ii) middle of page 22 - P_1(omega_1) is not defined (is it the L1 norm or other sparsity inducing penalty?)

Reviewer #2: In this paper the authors discuss the problem of availability of the most relevant tissue in the context of transcriptome-wide association study. They propose a “TWAS using sparse canonical correlation analysis” approach, and demonstrate performance through simulation studies and real data application for various traits. Overall, this is an excellent manuscript. It is clearly written, logically built up and the method is solid. The simulation study is well-conducted and the real example is reasonable. There are a few issues I’d like to highlight in order to improve the manuscript.

Major comments.

1. In sCCA-TWAS, the authors propose to use canonical variates Xu_1, Xu_2, and Xu_3 as the repeated measure of gene expression across tissue. Say Gv_1 is the corresponding canonical variate. The canonical correlation cor(Xu_1, Gv_1) is maximized, but it is not necessarily true that cor(Xu_1, G) is also maximized. How would the authors validate the choice of this particular linear combination of gene expressions across tissues instead of others?

2. In the real data application, the authors report no novel discoveries or indications of previous results. For completeness, what biology has this new application revealed? Please discuss in the Discussion.

Minor comments.

Line 404. The authors might mean “tissues corresponding to the columns”? so that consistent with Line 459.

Line 409 & 415. Subscript missing in “omega”.

Line 418. What are X_1 and X_2? Not defined.

Line 428. Are c1 and c2 chosen at each round or only for the first canonical variate pair?

Line 478. What are p and q?

Legends of the figure could be improved, so that it’s easier for readers to get the information.

Reviewer #3: The authors proposed to use sparse canonical correlation analysis (sCCA) to construct cross-tissue features, and then combine both single-tissue TWAS and sCCA TWAS results by an aggregate Cauchy association test (ACAT) for TWAS. The authors showed that this sCCA-ACAT has the highest power in their simulation studies and identified the greatest number of significant genes in their real TWAS of 10 complex phenotypes. I have the following major comments:

1. I think the most comparable TWAS method the authors need to include in their comparison is the UTMOST instead of single tissue GBJ. Even though the UTMOST also used GBJ approach to aggregate test across multiple tissue types, the main feature of UTMOST is that it actually estimates TWAS weights jointly across multiple tissue types.

2. As shown in the simulation studies, using sCCA alone actually had the lowest power in the simulation studies, while combining sCCA and single tissue TWAS by ACAT gave the highest power. The authors might also be better to include Single Tissue ACAT into comparison to show that there will still be improvement for aggregating sCCA and single tissue TWAS, with the confounding factor of aggregating method ACAT excluded.

3. The authors mentioned Bonferroni correction for multiple tests in their TWAS comparison studies. Actually, it would worth to consider the number of independent test for Bonferroni correction, instead of taking 0.05/(number of tests) as the significance threshold. Otherwise, the Bonferroni correction will lead to conservative results, especially when the considered gene expression levels are correlated across multiple tissue types.

4. As shown in S4 Fig, there is a huge inflation for testing significant heritability by sCCA method. I do not think showing the “power” for testing significant heritability in Fig 2 make any sense.

5. For type I error evaluation of the TWAS method, the authors should consider using the genome-wide gene-based association study threshold 2.5e-6 as the significance level, instead of current 0.05.

6. Even though the sCCA/PCA approach provides some weights/coefficients for the gene expression of multiple tissue types, the sensitivity and specificity to identify the true relevant tissue type is around 0.5 and 0.7, with expression heritability 10% and correlation 0.3 with the causal tissue. This shows that it might still be challenging to identify the true causal tissue type?

7. Besides providing a greater number of “significant” genes, the approach of testing the association between cis-SNPs and the sCCA/PCA features needs more justification in the perspective of biology. Does more mean higher power? What dose such a significant association mean?

8. Table 1 can be improved by using footnote to denote Reference papers and simplify text headers.

9. Figure 1 is hard to read. The authors may either split the flow chart per method, or only make one for the sCCA-ACAT method.

10. The lines in Fig 3 are hard to see.

11. Figures 6, 7, 8 could be combined as 3 panels in one figure.

12. All texts in the figures are hard to read.

**Have all data underlying the figures and results presented in the manuscript been provided?**

Reviewer #1: Yes

Reviewer #2: Yes

Reviewer #3: Yes

PLOS authors have the option to publish the peer review history of their article (what does this mean?). If published, this will include your full peer review and any attached files.

Reviewer #1: No

Reviewer #2: No

Reviewer #3: No

---

## [Decision Letter · Decision Letter 1]

6 Jan 2021

Dear Dr Kraft,

Thank you very much for submitting your Research Article entitled 'Leveraging expression from multiple tissues using sparse canonical correlation analysis (sCCA) and aggregate tests improves the power of transcriptome-wide association studies (TWAS)' to PLOS Genetics.

The revised manuscript was fully evaluated at the editorial level and by three independent peer reviewers. While two of the reviewers were satisfied with the revision, one reviewer continued to express several concerns with the manuscript. After editorial discussions, we have decided to allow you an opportunity to submit a revised manuscript that must fulfill the following requirements:

1) Authors must confirm uploading of all sCCA cross-tissue feature weights into the TWAS HUB web interface 

2) Authors must provide a user manual and provide more detailed code documentation for their sCCA-TWAS software to enable the tool to be used by the broad genetics community 

Additionally, it is recommended that the authors apply UTMOST to their real data example in Table 1 and compare and contrast the number of significant genes identified by this existing method compared to the proposed methodology.

[LINK]

Yours sincerely,

Michael P. Epstein

Associate Editor

PLOS Genetics

David Balding

Section Editor: Methods

PLOS Genetics

Reviewer's Responses to Questions

**Comments to the Authors:**

Reviewer #1: The response to some of my comments has been unsatisfactory. Without an efficient method, software and a database of feature, there is no clear advance to the field described in this manuscript.

On software and database: the authors talk a plan for sharing but there is no concrete database. Also what is shown on the GitHub page is not software, but just an example code.

On sCCA-TWAS methodology: the missing data is an important component of this and just adding something to the discussion does not answer the issues I raised. The drop in efficiency when using this un-validated imputation method is substantial.

Reviewer #3: All of my comments are addressed. Paper is well written and clear. Good to see such a useful method/pipeline available for TWAS.

Reviewer #4: The paper is very well written and the analyses well described. The authors have done a very good revision from their earlier version. The authors took into account my comments and they addresses all of them. I have no further suggestions.

**Have all data underlying the figures and results presented in the manuscript been provided?**

Reviewer #1: Yes

Reviewer #3: Yes

Reviewer #4: Yes

PLOS authors have the option to publish the peer review history of their article (what does this mean?). If published, this will include your full peer review and any attached files.

Reviewer #1: No

Reviewer #3: No

Reviewer #4: No

---

## [Editor Report · Decision Letter 2]

16 Mar 2021

Dear Pete,

We are pleased to inform you that your manuscript entitled "Leveraging expression from multiple tissues using sparse canonical correlation analysis and aggregate tests improves the power of transcriptome-wide association studies" has been editorially accepted for publication in PLOS Genetics. Congratulations!

Yours sincerely,

Michael P. Epstein

Associate Editor

PLOS Genetics

David Balding

Section Editor: Methods

PLOS Genetics

Comments from the reviewers (if applicable):

**Data Deposition**

http://datadryad.org/submit?journalID=pgenetics&manu=PGENETICS-D-20-01011R2

**Press Queries**

---

## [Editor Report · Acceptance letter]

31 Mar 2021

PGENETICS-D-20-01011R2 

Leveraging expression from multiple tissues using sparse canonical correlation analysis and aggregate tests improves the power of transcriptome-wide association studies 

Dear Dr Kraft, 

We are pleased to inform you that your manuscript entitled "Leveraging expression from multiple tissues using sparse canonical correlation analysis and aggregate tests improves the power of transcriptome-wide association studies" has been formally accepted for publication in PLOS Genetics! Your manuscript is now with our production department and you will be notified of the publication date in due course.

With kind regards,

Katalin Szabo

PLOS Genetics

On behalf of:
